# Early analysis of the Australian COVID-19 epidemic

**David J Price**[1,2†*], **Freya M Shearer**[1†*], **Michael T Meehan**[3], **Emma McBryde**[3], **Robert Moss**[1], **Nick Golding**[4], **Eamon J Conway**[2], **Peter Dawson**[5], **Deborah Cromer**[6,7], **James Wood**[8], **Sam Abbott**[9], **Jodie McVernon**[1,2,10], **James M McCaw**[1,2,11]

[1]Centre for Epidemiology and Biostatistics, Melbourne School of Population and Global Health, The University of Melbourne, Melbourne, Australia; [2]Victorian Infectious Diseases Reference Laboratory Epidemiology Unit at The Peter Doherty Institute for Infection and Immunity, The University of Melbourne and Royal Melbourne Hospital, Melbourne, Australia; [3]Australian Institute of Tropical Health and Medicine, James Cook University, Townsville, Australia; [4]Telethon Kids Institute and Curtin University, Perth, Australia; [5]Defence Science and Technology, Department of Defence, Canberra, Australia; [6]Kirby Institute for Infection and Immunity, University of New South Wales, Sydney, Australia; [7]School of Mathematics and Statistics, University of New South Wales, Sydney, Australia; [8]School of Public Health and Community Medicine, University of New South Wales, Sydney, Australia; [9]Centre for the Mathematical Modelling of Infectious Diseases, Department of Infectious Disease Epidemiology, London School of Hygiene and Tropical Medicine, London, United Kingdom; [10]Infection and Immunity Theme, Murdoch Children's Research Institute, The Royal Children's Hospital, Melbourne, Australia; [11]School of Mathematics and Statistics, The University of Melbourne, Melbourne, Australia

**\*For correspondence:**
david.price1@unimelb.edu.au
(DJP);
freya.shearer@unimelb.edu.au
(FMS)

[†]These authors contributed
equally to this work

**Competing interest:** See
page 11

**Reviewing editor:** Ben S
Cooper, Mahidol University,
Thailand

**Abstract** As of 1 May 2020, there had been 6808 confirmed cases of COVID-19 in Australia. Of these, 98 had died from the disease. The epidemic had been in decline since mid-March, with 308 cases confirmed nationally since 14 April. This suggests that the collective actions of the Australian public and government authorities in response to COVID-19 were sufficiently early and assiduous to avert a public health crisis – for now. Analysing factors that contribute to individual country experiences of COVID-19, such as the intensity and timing of public health interventions, will assist in the next stage of response planning globally. We describe how the epidemic and public health response unfolded in Australia up to 13 April. We estimate that the effective reproduction number was likely below one in each Australian state since mid-March and forecast that clinical demand would remain below capacity thresholds over the forecast period (from mid-to-late April).

## Introduction

A small cluster of cases of the disease now known as COVID-19 was first reported on December 29, 2019, in the Chinese city of Wuhan (*World Health Organization, 2020a*). By early May 2020, the disease had spread to all global regions, and overwhelmed some the world's most developed health systems. More than 2.8 million cases and 260,000 deaths had been confirmed globally, and the vast majority of countries with confirmed cases were reporting escalating transmission (*World Health Organization, 2020b*).

As of 1 May 2020, there were 6808 confirmed cases of COVID-19 in Australia. Of these, 98 had died from the disease. Encouragingly, the daily count of new confirmed cases had been declining since late March 2020, with 308 cases reported nationally since 14 April (*Australian Government Department of Health, 2020a*). This suggests that Australia has (to date) avoided a "worst-case" scenario — one where planning models estimated a peak daily demand for 35,000 ICU beds by around May 2020, far exceeding the health system's capacity of around 2,200 ICU beds (*Moss et al., 2020*).

The first wave of COVID-19 epidemics, and the government and public responses to them, have varied vastly across the globe. For example, many European countries and the United States are in the midst of explosive outbreaks with overwhelmed health systems (*Remuzzi and Remuzzi, 2020*; *The Lancet, 2020*). Meanwhile, countries such as Singapore and South Korea had early success in containing the spread, partly attributed to their extensive surveillance efforts and case targeted interventions (*Ng et al., 2020*; *COVID-19 National Emergency Response Center, Epidemiology and Case Management Team, Korea Centers for Disease Control and Prevention, 2020*). However, despite those early successes, Singapore has recently taken additional steps to further limit transmission in the face of increasing importations and community spread (*Government of Singapore, 2020*). Other locations in the region, including Taiwan, Hong Kong and New Zealand, have had similar epidemic experiences, achieving control through a combination of border, case targeted and social distancing measures.

Analysing key epidemiological and response factors — such as the intensity and timing of public health interventions — that contribute to individual country experiences of COVID-19 will assist in the next stage of response planning globally.

Here we describe the course of the COVID-19 epidemic and public health response in Australia from 22 January up to mid-April 2020 (summarised in *Figure 1*). We then quantify the impact of the public health response on disease transmission (*Figure 2*) and forecast the short-term health system demand from COVID-19 patients (*Figure 3*).

## Timeline of the Australian epidemic

Australia took an early and precautionary approach to COVID-19. On 1 February, when China was the only country reporting uncontained transmission, Australian authorities restricted all travel from mainland China to Australia, in order to reduce the risk of importation of the virus. Only Australian citizens and residents (and their dependants) were permitted to travel from China to Australia. These individuals were advised to self-quarantine for 14 days from their date of arrival. Further border measures, including enhanced testing and provision of additional advice, were placed on arrivals from other countries, based on a risk-assessment tool developed in early February (*Shearer et al., 2020*).

The day before Australia imposed these restrictions (January 31), 9720 cases of COVID-19 had been reported in mainland China (*World Health Organization, 2020c*). Australia had so far detected and managed nine imported cases, all with recent travel history from or a direct epidemiological link to Wuhan (*Australian Government Department of Health, 2020b*). Before the restrictions, Australia was expecting to receive approximately 200,000 air passengers from mainland China during February 2020 (*Australian Bureau of Statistics, 2019*). Travel numbers fell dramatically following the imposed travel restrictions.

These restrictions were not intended (and highly unlikely [*Errett et al., 2020*]) to prevent the ultimate importation of COVID-19 into Australia. Their purpose was to delay the establishment of an epidemic, buying valuable time for health authorities to plan and prepare.

During the month of February, with extensive testing and case targeted interventions (case isolation and contact quarantine) initiated from 29 January (*Australian Government Department of Health, 2020d*), Australia detected and managed only 12 cases. Meanwhile, globally, the geographic extent of transmission and daily counts of confirmed cases and deaths continued to increase drastically (*World Health Organization, 2020d*). In early March, Australia extended travel restrictions to a number of countries with large uncontained outbreaks, namely Iran (as of 1 March) (*Commonwealth Government of Australia, 2020a*), South Korea (as of 5 March) (*Commonwealth Government of Australia, 2020b*) and Italy (as of 11 March) (*Commonwealth Government of Australia, 2020c*).

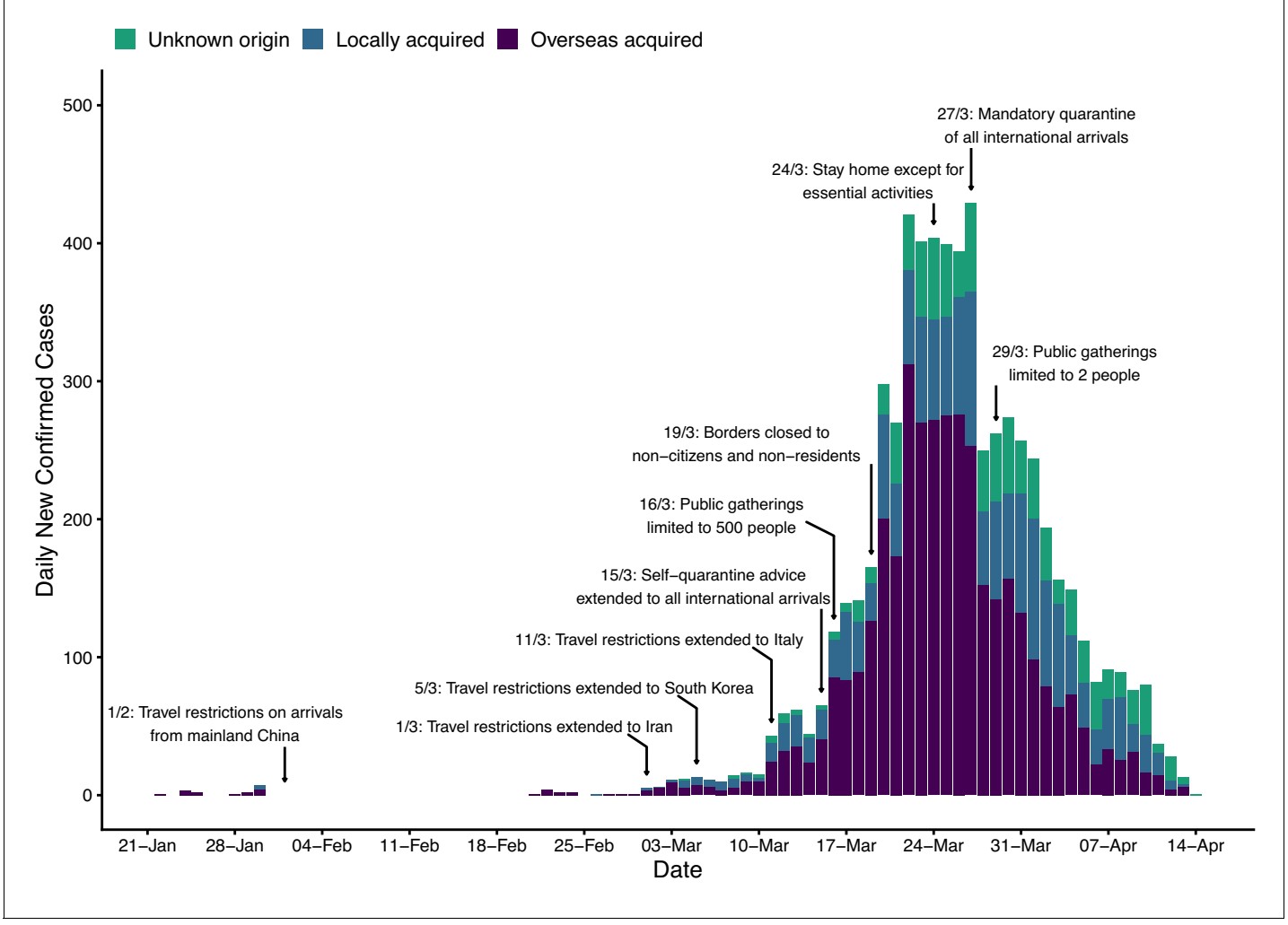

**Figure 1.** Time series of new daily confirmed cases of COVID-19 in Australia by import status (purple = overseas acquired, blue = locally acquired, green = unknown origin) from 22 January 2020 (first case detected) to 13 April 2020. Dates of selected key border and social distancing measures implemented by Australian authorities are indicated by annotations above the plotted case counts. These measures were in addition to case targeted interventions (case isolation and contact quarantine) and further border measures, including enhanced testing and provision of advice, on arrivals from other selected countries, based on a risk-assessment tool developed in early February (*Shearer et al., 2020*). Note that Australian citizens and residents (and their dependants) were exempt from travel restrictions, but upon returning to Australia were required to quarantine for 14 days from the date of arrival. A full timeline of social distancing and border measures is provided in *Figure 1—figure supplement 2*.
The online version of this article includes the following figure supplement(s) for figure 1:

**Figure supplement 1.** Time series of new daily confirmed cases of COVID-19 in each Australian state/territory by import status (purple = overseas acquired, blue = locally acquired, green = unknown origin) from 22 January 2020 (first case detected) to 13 April 2020.
**Figure supplement 2.** Timeline of border and social distancing measures implemented in Australia up to 4 April 2020.

Despite these measures, the daily case counts rose sharply in Australia during the first half of March. While the vast majority of these cases were connected to travellers returning to Australia from overseas, localised community transmission had been reported in areas of Sydney (NSW) and Melbourne (VIC) (*Australian Government Department of Health, 2020c*). Crude plots of the cumulative number of cases by country showed Australia on an early trajectory similar to the outbreaks experienced in China, Europe and the United States, where health systems had become or were becoming overwhelmed (*Australian Government Department of Health, 2020f*).

From 16 March, the Australian Government progressively implemented a range of social distancing measures in order to reduce and prevent further community transmission (*Commonwealth Government of Australia, 2020d*). The day before, authorities had imposed a

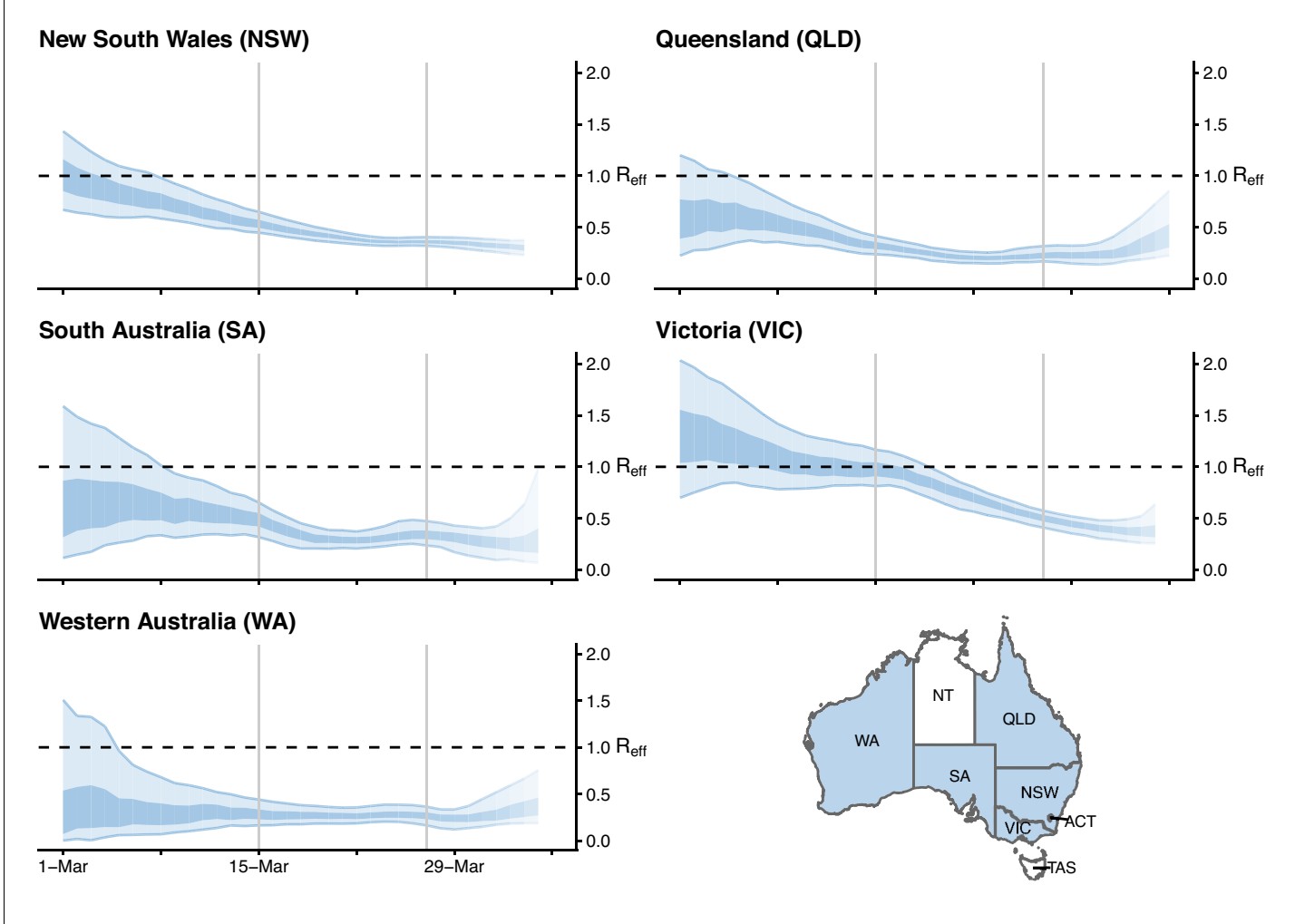

**Figure 2.** Time-varying estimate of the effective reproduction number ($R_{eff}$) of COVID-19 by Australian state (light blue ribbon = 90% credible interval; dark blue ribbon = 50% credible interval) from 1 March to 5 April 2020, based on data up to and including 13 April 2020. Confidence in the estimated values is indicated by shading with reduced shading corresponding to reduced confidence. The horizontal dashed line indicates the target value of 1 for the effective reproduction number required for control. Not presented are the Australian Capital Territory (ACT), Northern Territory (NT) and Tasmania (TAS), as these states/territories had insufficient local transmission. The uncertainty in the $R_{eff}$ estimates represent variability in a population-level average as a result of imperfect data, rather than individual-level heterogeneity in transmission (*i.e.*, the variation in the number of secondary cases generated by each case).

The online version of this article includes the following figure supplement(s) for figure 2:

**Figure supplement 1.** Sensitivity analysis 1 of 3.
**Figure supplement 2.** Sensitivity analysis 2 of 3.
**Figure supplement 3.** Sensitivity analysis 3 of 3.

self-quarantine requirement on all international arrivals (***Commonwealth Government of Australia, 2020e***). On 19 March, Australia closed its borders to all non-citizens and non-residents (***Commonwealth Government of Australia, 2020f***), and on March 27, moved to a policy of mandatory quarantine for any returning citizens and residents (***Commonwealth Government of Australia, 2020g***). By 29 March, social distancing measures had been escalated to the extent that all Australians were strongly advised to leave their homes only for limited essential activities and public gatherings were limited to two people (***Commonwealth Government of Australia, 2020h***).

By late March, daily counts of new cases appeared to be declining, suggesting that these measures had successfully reduced transmission.

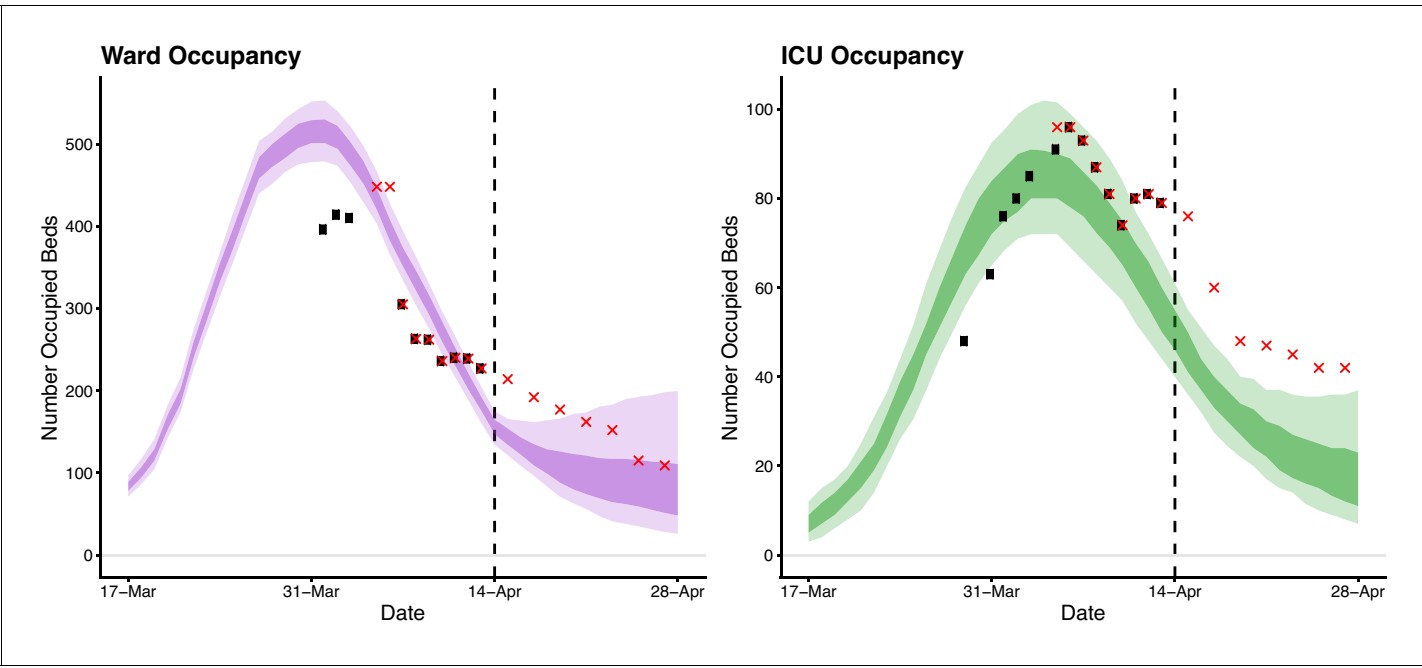

**Figure 3.** Forecasted daily hospital ward (left) and intensive care unit (right) occupancy (dark ribbons = 50% confidence intervals; light ribbons = 95% confidence intervals) from 17 March to 28 April. Occupancy = the number of beds occupied by COVID-19 patients on a given day. Black dots indicate the reported ward and ICU occupancy available from the Australian national COVID-19 database at the time. These data were retrospectively updated where complete data were available (red crosses). Australian health system ward and ICU bed capacities are estimated to be over 25,000 and 1,100, respectively, under the assumption that 50% of total capacity could possibly be dedicated to COVID-19 patients (*Australian Institute of Health and Welfare, 2019*). The forecasted daily case counts are shown in *Figure 3—figure supplement 1*.

The online version of this article includes the following figure supplement(s) for figure 3:

**Figure supplement 1.** Time series of new daily confirmed cases of COVID-19 in Australia from 1 March to 13 April 2020 (grey bars) overlaid by daily case counts estimated from the forecasting model up to April 13 and projected forward from 14 to 28 April inclusive.

## Quantifying the impact of the response

Quantifying changes in the rate of spread of infection over the course of an epidemic is critical for monitoring the collective impact of public health interventions and forecasting the short-term clinical burden. A key indicator of transmission in context is the effective reproduction number ($R_{eff}$) — the average number of secondary infections caused by an infected individual in the presence of public health interventions and for which no assumption of 100% susceptibility is made. If control efforts are able to bring $R_{eff}$ below 1, then on average there will be a decline in the number of new cases reported. The decline will become apparent after a delay of approximately one incubation period plus time to case detection and reporting following implementation of the control measure (*i.e.*, at least two weeks).

Using case counts from the Australian national COVID-19 database, we estimated $R_{eff}$ over time for each Australian state from 24 February to 5 April 2020 (*Figure 2*). We used a statistical method that estimates time-varying $R_{eff}$ by using an optimally selected moving average window (according to the continuous ranked probability score) to smooth the curve and reduce the impact of localised clusters and outbreaks that may cause large fluctuations (*London School of Hygiene & Tropical Medicine Mathematical Modelling of Infectious Diseases nCoV working group, 2020*). Importantly, the method accounts for time delays between illness onset and case notification. Incorporation of this lag is critical for accurate interpretation of the most recent data in the analysis, to be sure that an observed drop in the number of reported cases reflects an actual drop in case numbers.

Results show that $R_{eff}$ has likely been below one in each Australian state since early-to-mid March. These estimates are geographically averaged results over large areas and it is possible that $R_{eff}$ was much higher than one in a number of localised settings (see *Figure 2*). The estimated time-varying $R_{eff}$ value is based on cases that have been identified as a result of local transmission, whereas

imported cases only contribute to the force of infection. Imported and locally acquired cases were assumed to be equally infectious. The method for estimating $R_{eff}$ is sensitive to this assumption. Hence, we performed a sensitivity analysis to assess the impact of stepwise reductions in the infectiousness of imported cases on $R_{eff}$ as a result of quarantine measures implemented over time (see *Figure 2—figure supplement 1*, *Figure 2—figure supplement 2*, and *Figure 2—figure supplement 3*). The sensitivity analyses suggest that $R_{eff}$ may well have dropped below one later than shown in *Figure 2*.

In Victoria and New South Wales, the two Australian states with a substantial number of local cases, the effective reproduction number likely dropped from marginally above one to well below one within a two week period (considering both our main result and those from the sensitivity analyses) coinciding with the implementation of social distancing measures. A comparable trend was observed in New Zealand and many Western European countries, including France, Spain and Germany (*London School of Hygiene & Tropical Medicine Mathematical Modelling of Infectious Diseases nCoV working group, 2020*), where similar national, stage-wise social distancing policies were enacted (*Flaxman et al., 2020*). However, most of these European countries experienced widespread community transmission prior to the implementation of social distancing measures, with $R_{eff}$ estimates reaching between 1.5 and 2 in early March and declining over a longer period (three to four weeks) relative to Australia.

## Forecasting the clinical burden

Next we used our estimates of time-varying $R_{eff}$ to forecast the short-term clinical burden in Australia. Estimates were input into a mathematical model of disease dynamics that was extended to account for imported cases. A sequential Monte Carlo method was used to infer the model parameters and appropriately capture the uncertainty (*Moss et al., 2019a*), conditional on each of a number of sampled $R_{eff}$ trajectories up to 5 April, from which point they were assumed to be constant. The model was subsequently projected forward from April 14 to April 28, to forecast the number of reported cases, assuming a symptomatic detection probability of 80%.

The number of new daily hospitalisations and ICU admissions were estimated from recently observed and forecast case counts. Specifically, the age distribution of projected cases, and age-specific probabilities of hospitalisation and ICU admission, were extracted from Australian age-specific data on confirmed cases, assuming that this distribution would remain unchanged (see *Table 1*). In order to calculate the number of occupied ward/ICU beds per day, length-of-stay in a ward bed and ICU bed were assumed to be Gamma distributed with means (SD) of 11 (3.42) days and 14 (5.22) days, respectively. Our results indicated that with the public health interventions in place as of 13 April, Australia's hospital ward and ICU occupancy would remain well below capacity thresholds over the period from 14 to 28 April.

**Table 1.** Age-specific proportions of confirmed cases extracted from the Australian national COVID-19 database and age-specific estimates of the probability of hospitalisation and ICU admission for confirmed cases.

| Age | Proportion of cases | Pr(hospitalisation \| confirmed case) | Pr(ICU admission \| confirmed case) |
| --- | --- | --- | --- |
| 0-9 | 0.0102 | 0.1475 | 0.0000 |
| 10-18 | 0.0186 | 0.1081 | 0.0090 |
| 19-29 | 0.2258 | 0.0504 | 0.0007 |
| 30-39 | 0.1587 | 0.0865 | 0.0074 |
| 40-49 | 0.1291 | 0.0947 | 0.0208 |
| 50-59 | 0.1550 | 0.1112 | 0.0173 |
| 60-69 | 0.1686 | 0.1529 | 0.0318 |
| 70-79 | 0.1050 | 0.2440 | 0.0558 |
| 80+ | 0.0290 | 0.3815 | 0.0462 |

## Conclusions

Our analysis suggests that Australia's combined strategy of early, targeted management of the risk of importation, case targeted interventions, and broad-scale social distancing measures applied prior to the onset of (detected) widespread community transmission has substantially mitigated the first wave of COVID-19. More detailed analyses are required to assess the relative impact of specific response measures, and this information will be crucial for the next phase of response planning. Other factors, such as temperature, humidity and population density may influence transmission of SARS-CoV-2 (*Kissler et al., 2020*). Whether these factors have played a role in the relative control of SARS-CoV-2 in some countries, remains an open question. Noting that epidemics are established in both the northern and southern hemispheres, it may be possible to gain insight into such factors over the next six months, via for example a comparative analysis of transmission in Australia and Europe.

We further anticipated that the Australian health care system was well positioned to manage the projected COVID-19 case loads over the forecast period (up to 28 April). Ongoing situational assessment and monitoring of forecast hospital and ICU demand will be essential for managing possible future relaxation of broad-scale community interventions. Vigilance for localised increases in epidemic activity and in particular for outbreaks in vulnerable populations such as residential aged care facilities, where a high proportion of cases are likely to be severe, must be maintained.

One largely unknown factor at present is the proportion of SARS-CoV-2 infections that are asymptomatic, mild or undiagnosed. Even if this number is high, the Australian population would still be largely susceptible to infection. Accordingly, complete relaxation of the measures currently in place would see a rapid resurgence in epidemic activity. This problem is not unique to Australia. Many countries with intensive social distancing measures in place are starting to grapple with their options and time frames for a gradual return to relative normalcy (*Gottlieb et al., 2020*).

There are difficult decisions ahead for governments, and for now Australia is one of the few countries fortunate enough to be able to plan the next steps from a position of relative calm as opposed to crisis.

## Materials and methods

### Estimating the time-varying effective reproduction number

#### Overview

The method used to estimate $R_{eff}$ is described in *Cori et al., 2013*, as implemented in the `R package`, `EpiNow` (*Abbott et al., 2020*). This method is currently in development by the Centre for the Mathematical Modelling of Infectious Diseases at the London School of Hygiene and Tropical Medicine (*London School of Hygiene & Tropical Medicine Mathematical Modelling of Infectious Diseases nCoV working group, 2020*). Full details of their statistical analysis and code base is available via their website (https://epiforecasts.io/covid/).

The uncertainty in the $R_{eff}$ estimates (shown in *Figure 2*; *Figure 2—figure supplements 1*, *2* and *3*) represents variability in a population-level average as a result of imperfect data, rather than individual-level heterogeneity in transmission (*i.e.*, the variation in the number of secondary cases generated by each case). This is akin to the variation represented by a confidence interval (*i.e.*, variation in the estimate resulting from a finite sample), rather than a prediction interval (*i.e.*, variation in individual observations).

We provide a brief overview of the method and sources of imperfect data below, focusing on how the analysis was adapted to the Australian context.

#### Data

We used line-lists of reported cases for each Australian state/territory extracted from the national COVID-19 database. The line-lists contain the date when the individual first exhibited symptoms, date when the case notification was received by the jurisdictional health department and where the infection was acquired (*i.e.*, overseas or locally).

## Reporting delays and under-reporting

A *pre-hoc* statistical analysis was conducted in order to estimate a distribution of the reporting delays from the line-lists of cases, using the code base provided by *London School of Hygiene & Tropical Medicine Mathematical Modelling of Infectious Diseases nCoV working group, 2020*. The estimated reporting delay is assumed to remain constant over time. These reporting delays are used to: (i) infer the time of symptom onset for those without this information, and; (ii) infer how many cases in recent days are yet to be recorded. Adjusting for reporting delays is critical for inferring when a drop in observed cases reflects a true drop in cases.

Trends identified using this approach are robust to under-reporting, assuming that it is constant. However, absolute values of $R_{eff}$ may be biased by reporting rates. Pronounced changes in reporting rates may also impact the trends identified.

The delay from symptom onset to reporting is likely to decrease over the course of the epidemic, due to improved surveillance and reporting. We used a delay distribution estimated from observed reporting delays from the analysis period, which is therefore likely to underestimate reporting delays early in the epidemic, and overestimate them as the epidemic progressed. Underestimating the delay would result in an overestimate of $R_{eff}$, as the inferred onset dates (for those that were unknown) and adjustment for right-truncation, would result in more concentrated inferred daily cases (*i.e.*, the inferred cases would be more clustered in time than in reality). The converse would be true when overestimating the delay. The impact of this misspecified distribution will be greatest on the most recent estimates of $R_{eff}$, where inference for both right-truncation and missing symptom onset dates is required.

## Estimating the effective reproduction number over time

Briefly, the $R_{eff}$ was estimated for each day from 24 February 2020 up to 5 April 2020 using line list data – date of symptom onset, date of report, and import status – for each state. The method assumes that the serial interval (*i.e.*, time between symptom onset for an index and secondary case) is uncertain, with a mean of 4.7 days (95% CrI: 3.7, 6.0) and a standard deviation of 2.9 days (95% CrI: 1.9, 4.9), as estimated from early outbreak data in Wuhan, China (*Nishiura et al., 2020*). Combining the incidence over time with the uncertain distribution of serial intervals allows us to estimate $R_{eff}$ over time.

A different choice of serial interval distribution would affect the estimated time varying $R_{eff}$. This sensitivity is explored in detail in *Flaxman et al., 2020*, though we provide a brief description of the impact here. For the same daily case data, a longer average serial interval would correspond to an increased estimate of $R_{eff}$ when $R_{eff} > 1$, and a decreased estimate when $R_{eff} < 1$. This effect can be understood intuitively by considering the epidemic dynamics in these two situations. When $R_{eff} > 1$, daily case counts are increasing on average. The weighted average case counts (weighted by the serial interval distribution), decrease as the mean of the serial interval increases (*i.e.*, as the support is shifted to older/lower daily case data). In order to generate the same number of observed cases in the present, $R_{eff}$ must increase. A similar observation can be made for $R_{eff} < 1$.

In the context of our analyses (*Figure 2*), when the estimated $R_{eff}$ is above 1, assuming a longer mean serial interval would further increase the $R_{eff}$ estimates in each jurisdiction (*i.e.*, the upper 75% of the Victorian posterior distribution for approximately the first 7–10 days, while stretching the upper tails in the other jurisdictions). When the estimated $R_{eff}$ is below 1, a higher mean serial interval would further decrease those estimates. Qualitatively, this does not impact on the time series of $R_{eff}$ in each Australian jurisdiction.

A prior distribution was specified for $R_{eff}$, with mean 2.6 (informed by *Imai et al., 2020*) and a broad standard deviation of 2 so as to allow for a range of $R_{eff}$ values. Finally, $R_{eff}$ is estimated with a moving average window, selected to optimise the continuous ranked probability score, in order to smooth the curve and reduce the impact of localised events (*i.e.*, cases clustered in time) causing large variations.

Note that up to 20% of reported cases in the Australian national COVID-19 database do not have a reported import status (see *Figure 1*). Conservatively, we assumed that all cases with an unknown or unconfirmed source of acquisition were locally acquired.

## Accounting for imported cases

A large proportion of cases reported in Australia from January until now were imported from overseas. It is critical to account for two distinct populations in the case notification data – imported and locally acquired – in order to perform robust analyses of transmission in the early stages of this outbreak. The estimated time-varying $R_{eff}$ value is based on cases that have been identified as a result of local transmission, whereas imported cases contribute to transmission only (*Thompson et al., 2019*).

Specifically, the method assumes that local and imported cases contribute equally to transmission. The results under this assumption are presented in *Figure 2*. However, it is likely that imported cases contributed relatively less to transmission than locally acquired cases, as a result of quarantine and other border measures which targeted these individuals (*Figure 1—figure supplement 2*). In the absence of data on whether the infector of local cases was themselves an imported or local case (from which we could robustly estimate the contribution of imported cases to transmission), we explored this via a sensitivity analysis. We aimed to explore the impact of a number of plausible scenarios, based on our knowledge of the timing, extent and level of enforcement of different quarantine policies enacted over time.

Prior to 15 March, returning Australian residents and citizens (and their dependents) from mainland China were advised to self-quarantine. Note that further border measures were implemented during this period, including enhanced testing and provision of advice on arrivals from selected countries based on a risk assessment tool developed in early February (*Shearer et al., 2020*). On 15 March, Australian authorities imposed a *self-quarantine* requirement on all international arrivals, and from 27 March, moved to a *mandatory* quarantine policy for all international arrivals.

Hence for the sensitivity analysis, we assumed two step changes in the effectiveness of quarantine of overseas arrivals (timed to coincide with the two key policy changes), resulting in three intervention phases: prior to 15 March (self-quarantine of arrivals from selected countries); 15–27 March inclusive (self-quarantine of arrivals from all countries); and 27 March onward (mandatory quarantine of overseas arrivals from all countries). We further assumed that the relative infectiousness of imported cases decreased with each intervention phase. The first two intervention phases correspond to self-quarantine policies, so we assume that they resulted in a relatively small reduction in the relative infectiousness of imported cases (the first smaller than the second, since the pre-15 March policy only applied to arrivals from selected countries). The third intervention phase corresponds to *mandatory* quarantine of overseas arrivals in hotels which we assume is highly effective at reducing onward transmission from imported cases, but allows for the occasional transmission event. We then varied the percentage of imported cases contributing to transmission over the three intervention phases, as detailed in *Table 2*.

## Forecasting short-term ward and ICU bed occupancy

We used the estimates of time-varying $R_{eff}$ to forecast the national short-term ward/ICU occupancy due to COVID-19 patients.

**Table 2.** Percentage of imported cases assumed to be contributing to transmission over three intervention phases for each sensitivity analysis.

We assume two step changes in the effectiveness of quarantine of overseas arrivals, resulting in three intervention phases: prior to 15 March (self-quarantine of arrivals from selected countries); 15–27 March inclusive (self-quarantine of arrivals from all countries); and 27 March onward (mandatory quarantine of overseas arrivals from all countries).

| | Imported cases contributing to transmission | | |
| --- | --- | --- | --- |
| Sensitivity analysis | Prior to 15 March | 15–27 March | 27 March– |
| 1 | 90% | 50% | 1% |
| 2 | 80% | 50% | 1% |
| 3 | 50% | 20% | 1% |

The results of these three analyses are shown in **Figure 2—figure supplements 1**, **2** and **3**, respectively.

## Forecasting case counts

The forecasting method combines an SEEIIR (susceptible-exposed-infectious-recovered) population model of infection with daily COVID-19 case notification counts, through the use of a bootstrap particle filter (*Arulampalam et al., 2002*). This approach is similar to that implemented and described in *Moss et al., 2019b*, in the context of seasonal influenza forecasts for several major Australian cities. Briefly, the particle filter method uses post-regularisation (*Doucet et al., 2001*), with a deterministic resampling stage (*Kitagawa, 1996*). Code and documentation are available at https://epifx.readthedocs.io/en/latest/. The daily case counts by date of diagnosis were modelled using a negative binomial distribution with a fixed dispersion parameter *k*, and the expected number of cases was proportional to the daily incidence of symptomatic infections in the SEEIIR model; this proportion was characterised by the observation probability. Natural disease history parameters were sampled from narrow uniform priors, based on values reported in the literature for COVID-19 (*Table 3*), and each particle was associated with an $R_{eff}$ trajectory that was drawn from the state/territory $R_{eff}$ trajectories in *Figure 2* up to 5 April, from which point they are assumed to be constant. The model was subsequently projected forward from April 14 to April 28, to forecast the number of reported cases, assuming a detection probability of 80%.

In order to account for imported cases, we used daily counts of imported cases to construct a time-series of the expected daily importation rate and, assuming that such cases were identified one week after initial exposure, introduced exposure events into each particle trajectory by adding an extra term to the force of infection equation.

Model equations below describe the flow of individuals in the population from the susceptible class (S), through two exposed classes ($E_1$, $E_2$), two infectious classes ($I_1$, $I_2$) and finally into a removed class (R). The state variables $S, E_1, E_2, I_1, I_2, R$ correspond to the proportion of individuals in the population (of size N) in each compartment. Given the closed population and unidirectional flow of individuals through the compartments, we evaluate the daily incidence of symptomatic individuals (at time *t*) as the change in cumulative incidence (the bracketed term in the expression for $\mathbb{E}[y_t]$ below). Two exposed and infectious classes are chosen such that the duration of time in the exposed or infectious period has an Erlang distribution. The corresponding parameters are given in *Table 2*.

Model equations:

$$
\begin{aligned}
\frac{dS}{dt} &= -\beta(t) \cdot S(I_1 + I_2) \\
\frac{dE_1}{dt} &= \beta(t) \cdot S(I_1 + I_2) - 2\sigma E_1 \\
\frac{dE_2}{dt} &= 2\sigma E_1 - 2\sigma E_2 \\
\frac{dI_1}{dt} &= 2\sigma E_2 - 2\gamma I_1 \\
\frac{dI_2}{dt} &= 2\gamma I_1 - 2\gamma I_2 \\
\frac{dR}{dt} &= 2\gamma I_2
\end{aligned}
$$

With initial conditions:

$$
\begin{aligned}
S(0) &= \frac{N-10}{N} \\
E_1(0) &= \frac{10}{N} \\
E_2(0) &= I_1(0) = I_2(0) = R(0) = 0
\end{aligned}
$$

Observation model:

$$
\begin{aligned}
\mathbb{E}[y_t] &= N \cdot p_{obs} \cdot [I_2(t) + R(t) - (I_2(t-1) + R(t-1))] \\
x_t &= [S(t), E_1(t), E_2(t), I_1(t), I_2(t), R(t), \beta^i(t), \sigma, \gamma, \tau] \\
\mathcal{L}(y_t \mid x_t) &\sim \text{NegBin}(\mathbb{E}[y_t], k)
\end{aligned}
$$

With time-varying transmission rate corresponding to $R_{eff}$ trajectory *i*:

$$
\beta^i(t) = \begin{cases} 0, & \text{if } t < \tau \\ R_{eff}^i(t) \cdot \gamma, & \text{if } t \geq \tau, \end{cases} \quad \text{for } i \in \{1, 2, ..., 10\}
$$

**Table 3.** SEEIIR forecasting model parameters.

| Parameter | Definition | Value/Prior distribution |
|---|---|---|
| σ | Inverse of the mean incubation period | $U(4^{-1}, 3^{-1})$ |
| γ | Inverse of the mean infectious period | $U(10^{-1}, 9^{-1})$ |
| τ | Time of first exposure (days since 2020-01-01) | $U(0, 28)$ |
| $p_{obs}$ | Probability of observing a case | 0.8 |
| k | Dispersion parameter on Negative-Binomial observation model | 100 |

## Forecasting ward and ICU bed occupancy from observed and projected case counts

The number of new daily hospitalisations and ICU admissions were estimated from recently observed and forecasted case counts by:

1. Estimating the age distribution of projected case counts using data from the national COVID-19 database on the age-specific proportion of confirmed cases;
2. Estimating the age-specific hospitalisation and ICU admission rates using data from the national COVID-19 database. We assumed that all hospitalisations and ICU admissions were either recorded or were missing at random (31% and 58% of cases had no information recorded under hospitalisation or ICU status, respectively);
3. Randomly drawing the number of hospitalisations/ICU admissions in each age-group (for both the observed and projected case counts) from a binomial distribution with number of trials given by the expected number of cases in each age group (from 1), and probability given by the observed proportion of hospitalisations/ICU admissions by age group (from 2).

Finally, in order to calculate the number of occupied ward/ICU beds per day, length-of-stay in a ward bed and ICU bed were assumed to be Gamma distributed with means (SD) of 11 (3.42) days and 14 (5.22) days, respectively. We assumed ICU admissions required a ward bed prior to, and following, ICU stay for a Poisson distributed number of days with mean 2.5. Relevant Australian data were not available to parameterise a model that captures the dynamics of patient flow within the hospital system in more detail. Instead, these distributions were informed by a large study of clinical characteristics of 1099 COVID-19 patients in China (*Guan et al., 2020*). This model provides a useful indication of hospital bed occupancy based on limited available data and may be updated as more specific data (*e.g.*, on COVID-19 patient length-of-stay) becomes available.

## Acknowledgements

This study represents surveillance data reported through the Communicable Diseases Network Australia (CDNA) as part of the nationally coordinated response to COVID-19. We thank public health staff from incident emergency operations centres in state and territory health departments, and the Australian Government Department of Health, along with state and territory public health laboratories. We thank members of CDNA for their feedback and perspectives on the study results. We thank Dr Jonathan Tuke for helping to assemble Australian national and state announcements of COVID-19 response measures.

## Additional information

### Competing interests

David J Price, Freya M Shearer, Michael T Meehan, Emma McBryde, Robert Moss, Nick Golding, Eamon J Conway, Peter Dawson, Deborah Cromer, James Wood, Sam Abbott, Jodie McVernon, James M McCaw: This work was undertaken with direct funding support from the Australian Government Department of Health, Office of Health Protection and has assisted the Australian Government in its epidemic response activities.

## Funding

| Funder | Author |
| --- | --- |
| Department of Health, Australian Government | James M McCaw |

The funders had no role in study design, data collection and interpretation, or the decision to submit the work for publication.

## Author contributions

David J Price, Conceptualization, Data curation, Software, Formal analysis, Validation, Visualization, Methodology, Writing - original draft, Writing - review and editing; Freya M Shearer, Conceptualization, Data curation, Formal analysis, Visualization, Methodology, Writing - original draft; Michael T Meehan, Emma McBryde, Nick Golding, Peter Dawson, Deborah Cromer, James Wood, Sam Abbott, Methodology, Writing - review and editing; Robert Moss, Data curation, Software, Formal analysis, Validation, Methodology, Writing - original draft; Eamon J Conway, Formal analysis, Writing - review and editing; Jodie McVernon, Supervision, Writing - review and editing; James M McCaw, Conceptualization, Formal analysis, Supervision, Funding acquisition, Visualization, Methodology, Writing - review and editing

## Author ORCIDs

David J Price https://orcid.org/0000-0003-0076-3123
Freya M Shearer https://orcid.org/0000-0001-9600-3473
James M McCaw http://orcid.org/0000-0002-2452-3098

## Decision letter and Author response

Decision letter https://doi.org/10.7554/eLife.58785.sa1
Author response https://doi.org/10.7554/eLife.58785.sa2

# Additional files

## Supplementary files

• Source code 1. Analysis code.

• Transparent reporting form

## Data availability

Analysis code is included in the supplementary materials. Datasets analysed and generated during this study are included in the supplementary materials. For estimates of the time-varying effective reproduction number (Figure 2), the complete line listed data within the Australian national COVID-19 database are not publicly available. However, we provide the cases per day by notification date and state (as shown in Figure 1 and Figure 1–figure supplement 1) which, when supplemented with the estimated distribution of the delay from symptom onset to notification (samples from this distribution are provided as a data file), analyses of the time-varying effective reproduction number can be performed.

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
