## [Decision Letter]

**Acceptance summary:**

This paper describe key features of the COVID-19 epidemic and public health response in Australia up until mid-April. It represents a concise and worthwhile contribution to the COVID-19 literature.

**Decision letter after peer review:**

Thank you for submitting your article "Early analysis of the Australian COVID-19 epidemic" for consideration by *eLife*. Your article has been reviewed by three peer reviewers, including Ben S Cooper as the Reviewing Editor and Reviewer #1, and the evaluation has been overseen by a Senior Editor. The following individual involved in review of your submission has agreed to reveal their identity: Andrew James Kerr Conlan (Reviewer #3).

The reviewers have discussed the reviews with one another and the Reviewing Editor has drafted this decision to help you prepare a revised submission. Please aim to submit a revised version that addresses these concerns directly. Although we expect that you will address these comments in your response letter we also need to see the corresponding revision in the text of the manuscript. Some of the reviewers' comments may seem to be simple queries or challenges that do not prompt revisions to the text. Please keep in mind, however, that readers may have the same perspective as the reviewers. Therefore, it is essential that you attempt to amend or expand the text to clarify the narrative accordingly.

Summary:

This paper describe key features of the COVID-19 epidemic and public health response in Australia up until mid-April. It represents a concise and worthwhile contribution to the COVID-19 literature. The paper mostly uses established methodologies (or recently described extensions of established methodology) and the findings are likely to be of broad interest. The main limitation (which is acknowledged by the authors) is that no attempt is made to quantify the likely effect of different interventions. Adding this would strengthen the paper but given that multiple measures were enacted at different times and the limited extent of the outbreak further investigation into the relative impact of different measures would be challenging and is not considered an essential revision.

Required revisions:

1) Subsection “Forecasting case counts”. The observation model is a little hard to understand and needs clarification. It appears to be saying that there are two points in the course of an infection when the infection might be observed (in both cases with probability *p_obs_*): either the day when the infected person enters the second infectious compartment (*I*_2_) or the day they cease to be infectious and enter the *R* compartment. With this formulation it looks like it's possible for there to be more observed infections than there are infected hosts. It's also not specified whether S, *E*_1_, *E*_2_ etc represent absolute numbers or the proportion of hosts in the compartments. The initial conditions suggest that these are proportions, but if they are proportions then the observation model makes no sense (as then the expected number of observed infections per day would be at most 1). It's also surprising that there is not a delay between the state transitions in the model and the observations *y_t_* (assuming *y_t_* represents the number of infections observed at time t). Maybe it's just that accounting for such a delay would make no practical difference to the conclusions and for that reason the delay can be ignored. If that's the case it's worth saying so. Also worth saying explicitly somewhere that units of time are days – this doesn't seem to be mentioned anywhere.

2) To put the findings into context it would be helpful to describe what was not done as well as what was. In particular, information on recommendations and practice regarding mask wearing and hand hygiene would be of interest as would information on use of/lack of use of contact tracing apps over this period. To put the results into context it might also be helpful to extend the Discussion to consider and contrast the magnitude of the estimated effective reproduction numbers before and after interventions compared to other countries if space permits.

3) "contact quarantine" is reported to have been used. It would be helpful to clarify the dates when this started and any information on the success and speed of tracing contacts of cases would be helpful here.

4) To help put the Australian experience in perspective it would also be helpful to briefly give information on factors that might influence spread (such as temperature, humidity, crowding etc) and perhaps consider these (very briefly) in the Discussion.

5) Subsection “Forecasting the clinical burden”: Please give a brief explanation of the sequential Monte Carlo method used in the Materials and methods section.

6) Subsection “Estimating the effective reproduction number over time”: "optimally selected". Selected to optimise what?

7) Subsection “Accounting for imported cases”: "50%, 50% and 80%"?

8) "narrow uniform priors" – can these be specified in the Materials and methods.

9) Table 2: Can "Time of first exposure" be defined? Unclear what this means.

10) "Australia's symptomatic case ascertainment rate is very high (between 77 and 100%)".

This seems extremely high given that we know that many people experience very mild symptoms. Is this really credible? Unfortunately the link to London School of Hygiene and Tropical Medicine Mathematical Modelling of Infectious Diseases nCoV working group, 2020, which makes this claim does not seem to be working so it's not possible to confidently assess the assumptions behind this. However, if we assume the link should be to https://cmmid.github.io/topics/covid19/global_cfr_estimates.html this this would indicate that the estimate is based on the case fatality ratio, and the assumption that true CFR is 1.4% (based on Chinese data). However, case fatality ratio is highly age dependent and given lack of widespread dissemination in the community it seems at least possible that spread in Australia might have been largely confined to younger age groups leading to lower CFR. Given this estimate seems so surprising (and is not based on peer-reviewed research) it seems appropriate at least to add some caveats to this. Wouldn't this also depend on precisely what case definition is being used?

11) Figure 1—figure supplement 2 reports states "encouraging" parents to keep children from school. Is there any information that can be shared on what actually happened (i.e. what proportion of school aged children went to school)?

12) It would also be helpful to update the numbers reported in the Introduction.

13) Given that the time periods for model prediction are now in the past, it would be instructive to compare the predictions made with the actual numbers (or to provide some other form of model assessment).

14) Values for the serial interval were retrieved from early outbreak data in Wuhan. There are more recent estimations of the serial intervals now. What impact could this have on the results?

15) It is important to account for different infectivities of imported cases through sensitivity analyses. However justifications for the different percentages for the contribution to the transmission (Figure 2—figure supplements 1, 2, 3) is lacking. Are these arbitrary? It would be good to have an explanation.

16) It is not clear on which data or assumptions the parameters for the length-of-stay distribution in a ward or ICU bed are based on since no references are given.

17) It is surprising that there wasn't a change in time from onset to report during the outbreak. Could the authors clarify?

18) Could there be more detail on where the local transmissions occurred? Or is this reported elsewhere?

19) Could there be more clarity about the uncertainty presented in the *R* estimation figures legends? There seems to be a lot of confusion on Twitter etc in interpreting these graphs in terms of variance around the mean/median or variance in the *R* (some people transmitting more than others), therefore this is an opportunity to state very clearly what the uncertainty represents.

---

## [Author Response]

Summary:This paper describe key features of the COVID-19 epidemic and public health response in Australia up until mid-April. It represents a concise and worthwhile contribution to the COVID-19 literature. The paper mostly uses established methodologies (or recently described extensions of established methodology) and the findings are likely to be of broad interest. The main limitation (which is acknowledged by the authors) is that no attempt is made to quantify the likely effect of different interventions. Adding this would strengthen the paper but given that multiple measures were enacted at different times and the limited extent of the outbreak further investigation into the relative impact of different measures would be challenging and is not considered an essential revision.

We thank the reviewer for their positive comments. As well as providing a baseline for epidemic forecasts (which at the time of review are now being prepared), we agree that an additional direction in which to take this work is to quantify the likely effect of different interventions. As the reviewer has highlighted, this would be a challenging task, given the short timeframe in which multiple interventions were implemented and the lack of widespread community transmission. We believe that this work is beyond the scope of our *Short Report* which aims to provide a descriptive analysis of the early course of the Australian COVID-19 epidemic and public health response, as well as a quantitative analysis of the overall effectiveness of the public health response over this period.

Required revisions:1) Subsection “Forecasting case counts”. The observation model is a little hard to understand and needs clarification. It appears to be saying that there are two points in the course of an infection when the infection might be observed (in both cases with probability p_obs_): either the day when the infected person enters the second infectious compartment (I_2_) or the day they cease to be infectious and enter the R compartment. With this formulation it looks like it's possible for there to be more observed infections than there are infected hosts. It's also not specified whether S, E_1_, E_2_ etc represent absolute numbers or the proportion of hosts in the compartments. The initial conditions suggest that these are proportions, but if they are proportions then the observation model makes no sense (as then the expected number of observed infections per day would be at most 1). It's also surprising that there is not a delay between the state transitions in the model and the observations y_t_ (assuming y_t_ represents the number of infections observed at time t). Maybe it's just that accounting for such a delay would make no practical difference to the conclusions and for that reason the delay can be ignored. If that's the case it's worth saying so. Also worth saying explicitly somewhere that units of time are days – this doesn't seem to be mentioned anywhere.

The model captures the cases according to date of symptom onset, and this corresponds to the transition from *I_1_* into *I*_2_. Individuals are infectious in the population upon entry to *I*_1_, representing pre-symptomatic infectiousness. The formula in the observation model represents the daily incidence of symptomatic individuals. As a result of considering a closed population with unidirectional flow between compartments, this can be calculated as the change in cumulative incidence at time *t* via the formula: *I_2_*(*t*) + *R*(*t*) – (*I_2_*(*t* – 1) + *R*(*t* – 1)). We trust this explanation clarifies the basis for the model.

We thank the reviewers for highlighting the missing *N* in the observation model – the state variables *S*, *E*_1_, *E*_2_, *I*_1_, *I*_2_, *R* represent the proportion of individuals in each compartment, and the observation model should scale these values to the number of individuals. Note the omitted factor of *N* only occurred in the text. Our implementation in computer code includes the factor. We do not explicitly incorporate a reporting delay in this forecasting model. The model represents the number of individuals with a given symptom onset date. It indirectly incorporates reporting delays via the effective reproduction number estimates that are used in the forecast model, where *R_eff_* is estimated accounting for reporting delays.

In order to clarify these details, we have added the following text to the Materials and methods section:

“The state variables *S*, *E*_1_, *E*_2_, *I*_1_, *I*_2_, *R* correspond to the proportion of individuals in the population (of size *N*) in each compartment. Given the closed population and unidirectional flow of individuals through the compartments, we evaluate the daily incidence of symptomatic individuals (at time t) as the change in cumulative incidence (the bracketed term in the expression for 𝔼[*y_t_*] below).”

And, we have adjusted the formula for 𝔼[*y_t_*] to include the factor of *N*, and align the terms with the above description to:

𝔼[*y_t_*] = *N* · *p_obs_* · [*I*_2_(*t*) + *R*(*t*) – (*I*_2_(*t* – 1) + *R*(*t* – 1))]

2) To put the findings into context it would be helpful to describe what was not done as well as what was. In particular, information on recommendations and practice regarding mask wearing and hand hygiene would be of interest as would information on use of/lack of use of contact tracing apps over this period. To put the results into context it might also be helpful to extend the Discussion to consider and contrast the magnitude of the estimated effective reproduction numbers before and after interventions compared to other countries if space permits.

We thank the reviewer for these helpful suggestions.

The use of face masks by the general public was not recommended in Australia at any time during the analysis period (https://www.health.gov.au/resources/publications/coronavirus-covid-19-use-of-masks-by-the-public-in-the-community). Personal hygiene measures were recommended to the general public to slow the spread of SARS-CoV-2, including television, print, radio and social media campaigns commissioned by government. Contact tracing was performed by public health officials throughout the analysis period. A voluntary contact tracing app “COVIDSafe” was released later on 26 April (https://www.health.gov.au/ministers/the-hon-greg-hunt-mp/media/covidsafe-new-app-to-slow-the-spread-of-the-coronavirus).

We have added the following notes on these interventions to the caption of Figure 1—figure supplement 2:

“These measures were in addition to case targeted interventions, specifically case isolation and quarantine of their contacts. Contact tracing was initiated from 29 January 2020 and was performed by public health officials.”

And:

“Note 4: The use of face masks by the general public was not recommended at any time during the analysis period. Note 5: Personal hygiene measures and the “1.5m distancing rule" were promoted to the general public through television, print, radio and social media campaigns commissioned by national and state governments.”

In addition, we have added the following discussion to the end of the section “Quantifying the impact of the response” in the main text:

“In Victoria and New South Wales, the two Australian states with a substantial number of local cases, the effective reproduction number likely dropped from marginally above 1 to well below 1 within a two week period (considering both our main result and those from the sensitivity analyses) coinciding with the implementation of social distancing measures. […] However, most of these countries experienced widespread community transmission prior to the implementation of social distancing measures, with *R_eff_* estimates reaching between 1.5 and 2 in early March and declining over a longer period (three to four weeks) relative to Australia.”

3) "contact quarantine" is reported to have been used. It would be helpful to clarify the dates when this started and any information on the success and speed of tracing contacts of cases would be helpful here.

Unfortunately, there is no publicly available information on the effectiveness of contact tracing activities in Australia. Tracing of contacts of cases was initiated very early in the response, performed by public health units. From 29 January, individuals who had been in contact with any confirmed cases of COVID-19 were advised to quarantine in their home for 14 days following exposure. See statement by the Australian Health Protection Principal Committee: https://www.health.gov.au/news/australian-health-protection-principal-committee-ahppc-statement-on-novel-coronavirus-on-29-january-2020-0.

We have amended the section of the main text where case targeted interventions are first mentioned to indicate the date when these interventions were initiated and referenced the above statement by the AHPPC:

“During the month of February, with extensive testing and case targeted interventions (case isolation and contact quarantine) initiated from January 29 [Australian Government Department of Health, 2020], Australia detected and managed only 12 cases.”

4) To help put the Australian experience in perspective it would also be helpful to briefly give information on factors that might influence spread (such as temperature, humidity, crowding etc) and perhaps consider these (very briefly) in the Discussion.

We thank the reviewer for this important point. We now include some brief reflections on how these factors may influence spread in Australia in the Discussion:

[Existing text] “Our analysis suggests that Australia's combined strategy of early, targeted management of the risk of importation, case targeted interventions, and broad-scale social distancing measures applied prior to the onset of (detected) widespread community transmission has substantially mitigated the first wave of COVID-19. More detailed analyses are required to assess the relative impact of specific response measures, and this information will be crucial for the next phase of response planning.”

[Additional text] “Other factors, such as temperature, humidity and population density may influence transmission of SARS-CoV-2 [Gottlieb et al., 2020]. […] Noting that epidemics are now established in both the northern and southern hemispheres, it may be possible to gain insight into such factors over the next six months, via for example a comparative analysis of transmission in Australia and Europe.”

5) Subsection “Forecasting the clinical burden”: Please give a brief explanation of the sequential Monte Carlo method used in the Materials and methods section.

The Monte Carlo method used to first the forecasting model is a particle filter that is described in detail in the context of seasonal influenza forecasts in Moss et al., 2019. Briefly, the method uses post-regularisation as described in Doucet (10.1007/978-1-4757-3437-9), with a deterministic resampling stage as described in Kitagawa (10.1109/78.978374). The code and documentation are available at https: //epifx:readthedocs:io/en/latest/.

We have added the following text to the Materials and methods section:

“This approach is similar to that implemented and described in [Thompson et al., 2019], in the context of seasonal influenza forecasts for several major Australian cities. […] Code and documentation are available at https://epifx:readthedocs:io/en/latest/.”

6) Subsection “Estimating the effective reproduction number over time”: "optimally selected". Selected to optimise what?

The moving average window is chosen to optimise the continuous ranked probability score, a common metric to assess the forecasting ability of a model (here, how well variably smoothed estimates appropriately capture the estimated *R_eff_* on subsequent days).

We have amended the following sentence in the main text to include this detail:

“We used a statistical method that estimates time-varying *R_eff_* by using an optimally selected moving average window (according to the continuous ranked probability score) to smooth the curve and reduce the impact of localised clusters and outbreaks that may cause large fluctuations [London School of Hygiene and Tropical Medicine Mathematical Modelling of Infectious Diseases nCoV working group, 2020].”

And the following sentence in the Materials and methods section:

“Finally, *R_eff_* is estimated with a moving average window, selected to optimise the continuous ranked probability score, in order to smooth the curve and reduce the impact of localised events (i.e., cases clustered in time) causing large variations.”

7) Subsection “Accounting for imported cases”: "50%, 50% and 80%"?

We agree that the meaning of these different percentages and how they were derived was not clear in the original version of the manuscript. To improve clarity, we have made a number of additions and modifications to the Materials and methods section, which we describe in detail below, since another reviewer has raised a similar point.

8) "narrow uniform priors" – can these be specified in the Materials and methods.

The prior distributions are specified in Table 2, and we have added a reference to this table in the text.

9) Table 2: Can "Time of first exposure" be defined? Unclear what this means.

We apologise for the lack of clarity in the original version of the manuscript. The time of first exposure is the time at which exposure occurred for the first infected individuals (*i.e.*, entered *E*_1_). It represents the number of days since 1 January 2020. We have added this detail to the description in Table 2.

10) "Australia's symptomatic case ascertainment rate is very high (between 77 and 100%)".This seems extremely high given that we know that many people experience very mild symptoms. Is this really credible? Unfortunately the link to London School of Hygiene and Tropical Medicine Mathematical Modelling of Infectious Diseases nCoV working group, 2020, which makes this claim does not seem to be working so it's not possible to confidently assess the assumptions behind this. However, if we assume the link should be to https://cmmid.github.io/topics/covid19/global_cfr_estimates.html this this would indicate that the estimate is based on the case fatality ratio, and the assumption that true CFR is 1.4% (based on Chinese data). However, case fatality ratio is highly age dependent and given lack of widespread dissemination in the community it seems at least possible that spread in Australia might have been largely confined to younger age groups leading to lower CFR. Given this estimate seems so surprising (and is not based on peer-reviewed research) it seems appropriate at least to add some caveats to this. Wouldn't this also depend on precisely what case definition is being used?

The reviewer is correct that we are using the method described at the new web link. We agree with their assessment of the limitations of this method. As the reviewer has highlighted, estimates of the symptomatic case detection rate using this method are sensitive to age distributions and case definitions. Indeed, we are working with the group who produce these estimates to improve the method, including accounting for differences in population age-structure between Australia and China, where the baseline CFR was calculated (https://doi.org/10.1101/2020.07.07.20148460). These additional analyses revealed no substantial difference in the age-adjusted estimates for Australia. Further, a high case detection rate in Australia during the early phase of the epidemic is not unexpected, given that approximately two thirds of cases were overseas acquired, and quarantine and other border measures applied to the vast majority of these individuals. Hence, we would expect that, on average, cases were easier to detect than if the majority of cases were arising from community transmission.

We agree with the reviewer that these nuances and caveats should be described in the article where we refer to the estimated case detection rate. However, since our article is a *Short Report*, and we believe that the statement regarding the symptomatic case detection rate is not necessary for communicating our point, we have removed reference to it from the article. The relevant section of the amended paragraph is included below:

“One largely unknown factor at present is the proportion of SARS-CoV-2 infections that are asymptomatic, mild or undiagnosed. Even if this number is high, the Australian population would still be largely susceptible to infection. Accordingly, complete relaxation of the measures currently in place would see a rapid resurgence in epidemic activity.”

11) Figure 1—figure supplement 2 reports states "encouraging" parents to keep children from school. Is there any information that can be shared on what actually happened (i.e. what proportion of school aged children went to school)?

Unfortunately, there is no publicly available data on actual school attendance in response to government recommendations to keep children from school. A report prepared by the National Centre for Immunisation Research and Surveillance on COVID-19 in schools in the state of New South Wales (the largest (by population) state in Australia) reported that face-to-face attendance in schools decreased significantly after parents were encouraged to keep their children at home (http://ncirs:org:au/sites/default/files/2020-04/NCIRS%20NSW%20Schools%20COVID Summary FINAL%20public 26%20April%202020:pdf). Media reports from the state of Victoria (the second largest state) suggested that school attendance was substantially reduced nearly a week prior to the announcement by the state government that school holidays would be brought forward (https://www.theage.com.au/national/victoria/parents-are-voting-with-their-feet-school-attendancerates-in-freefall-20200317-p54aw0.html). The Education Minister of Victoria reported in the media that approximately 3% of students attended school on site on the first day of term on 16 April (https://www.abc.net.au/news/2020-04-16/victoria-reports-97-of-school-students-learning/12153628?nw=0).

We have added a note to the caption of Figure 1—figure supplement 2 to highlight these points:

“Note 3: School attendance is reported to have reduced substantially following government recommendations to keep children from school [Carey, 2020], and in some cases, prior to these announcements [Cori et al, 2013]. It should also be noted that school holidays in some states/territories overlapped with the restriction periods (late March and early April).”

12) It would also be helpful to update the numbers reported in the Introduction.

We have carefully considered this suggestion and decided to not change the dates and numbers in the Introduction. Our reasoning is that the manuscript presents an analysis of the Australian epidemic at the tail end of the first wave of epidemic activity. To adjust dates and numbers would have significant consequences for the entire manuscript, and shift the attention from an analysis of the early – and important – epidemic dynamics to a more contemporary analysis.

13) Given that the time periods for model prediction are now in the past, it would be instructive to compare the predictions made with the actual numbers (or to provide some other form of model assessment).

We have added actual data from the forecast period to our plots of projected daily case counts (Figure 3—figure supplement 1) and ward/ICU occupancy (Figure 3).

The ward and ICU occupancy forecasts were based on our early understanding of the dynamics, informed by occupancy data available at the time (shown as black dots). Noting that Australian data on ward/ICU length-of-stay and delay from symptom onset to admission were not available at the time. The occupancy data were updated retrospectively where complete data were available (red crosses) – providing a different perspective on the underlying dynamics. However, the model still provides a reasonable estimate of the projected ward and ICU occupancy over the two-week forecast period, in particular both forecasts capture the observed trend. We now use these updated data on hospitalisation, as well as age-structured information on hospital length-of-stay and symptom-onset-to-admission in the Australian context, in our ongoing occupancy forecasts.

14) Values for the serial interval were retrieved from early outbreak data in Wuhan. There are more recent estimations of the serial intervals now. What impact could this have on the results?

We thank the reviewers for highlighting this important detail. The reviewers are correct that the serial interval used was obtained from early Wuhan data reported in Nishiura et al., 2020. More recent estimates include serial intervals reported from Shenzhen (Bi et al.), or the generation interval distributions estimated for Singapore (Ganyani et al.) and Tianjin (Ganyani et al.). The estimated generation intervals (GI's) tend to have less variation — as these are only capturing the delay (and corresponding variation) in time from infection to infection, whereas the serial interval (SI) has greater variability as it must also capture the delay to infectiousness for both the infector and infectee.

The Singapore (GI) and Shenzhen (SI) estimates have marginally higher means (5.2, 6.3, respectively) than the chosen Nishiura SI estimate (4.7), whereas the Tianjin GI has marginally lower mean (3.95). A serial interval distribution with a larger mean (for the same case data) corresponds to an increased estimate of *R_eff_* when *R_eff_* >1, and a decreased estimate when *R_eff_* <1. These sensitivities are explored in detail in Flaxman et al., 2020 (both their Nature publication and the corresponding Report #13). When the epidemic is growing (i.e., *R_eff_*>1, and recent daily case counts are higher than the past), increasing the mean of the serial interval distribution will correspond to a reduced weighted sum of cases in the calculation of *R_eff_*– i.e., the effective “force of infection” will be lower – and so *R_eff_* must be larger in order to generate the same number of observed cases at present. The converse applies when *R_eff_*< 1, where lower recent daily case counts relative to the past will correspond to a larger effective force of infection in the *R_eff_* calculation, and so *R_eff_* must decrease to compensate. Where *R_eff_*= 1, changing the mean serial interval should not impact on the estimated *R_eff_*.

In order to briefly describe the effect of such a choice, we have added the following text to the Materials and methods section:

“A different choice of serial interval distribution would affect the estimated time varying *R_eff_*. […] Qualitatively, this does not impact on the time series of *R_eff_* in each Australian jurisdiction.”

15) It is important to account for different infectivities of imported cases through sensitivity analyses. However justifications for the different percentages for the contribution to the transmission (Figure 2—figure supplements 1, 2, 3) is lacking. Are these arbitrary? It would be good to have an explanation.

We agree that our rationale for the different percentages was not clear in the original version of the article. To improve clarity, we have made number of additions and modifications:

– We have substantially re-written the following explanation in the Materials and methods section:

“Specifically, the method assumes that local and imported cases contribute equally to transmission. —figure supplement[…] We then varied the percentage of imported cases contributing to transmission over the three intervention phases, as detailed in Table 1.”

– We now refer to the percentage of imported cases contributing to transmission rather than not contributing to transmission.

– Finally, we have now presented the assumptions of each sensitivity analysis in Table 1.

16) It is not clear on which data or assumptions the parameters for the length-of-stay distribution in a ward or ICU bed are based on since no references are given.

Thank you for drawing our attention to this oversight. In the absence of Australian data on hospital and ICU length-of-stay, we used estimates from a large study on the clinical characteristics of COVID-19 patients from China, Guan et al., 2020. We have now included this reference in the text.

17) It is surprising that there wasn't a change in time from onset to report during the outbreak. Could the authors clarify?

For the purposes of these analyses, it was assumed that the time from symptom onset to notification was constant. A time-varying estimate of the delay distribution did not feature in these analyses, though we have since incorporated this feature in our analyses and it is also now a feature of the LSHTM method of Abbott et al.

The delay from symptom onset to reporting is likely to decrease over the course of the epidemic, due to improved surveillance and reporting. We used a delay distribution estimated from observed reporting delays from the analysis period, which is therefore likely to underestimate reporting delays early in the epidemic and overestimate them as the epidemic progressed. Underestimating the delay would result in an overestimate of *R_eff_*, as the inferred onset dates (for those that were unknown) and adjustment for right-truncation, would result in more concentrated inferred daily cases (i.e., the inferred cases would be clustered on more recent dates than in reality). The converse would be true when overestimating the delay. The impact of this misspecified distribution will be greatest on the most recent estimates of *R_eff_*, where inference for both right-truncation and missing symptom onset dates is required.

We have added the following text to the Materials and methods section:

“The delay from symptom onset to reporting is likely to decrease over the course of the epidemic, due to improved surveillance and reporting. […] The impact of this misspecified distribution will be greatest on the most recent estimates of *R_eff_*, where inference for both right-truncation and missing symptom onset dates is required.”

18) Could there be more detail on where the local transmissions occurred? Or is this reported elsewhere?

Local transmission occurred in all Australian jurisdictions as shown in Figure 1—figure supplement 1, which displays a time series of cases for each jurisdiction coloured by acquisition status. Beyond reporting the time series of local case counts by state, the only localised community outbreaks, which occurred in Sydney (NSW) and Melbourne (VIC), are briefly mentioned in the main text. We have added a citation in the following line of the main text, indicating where details are available on the epidemiological characteristics of locally and overseas acquired infections:

“While the vast majority of these cases were connected to travellers returning to Australia from overseas, localised community transmission had been reported in areas of Sydney (NSW) and Melbourne (VIC) [Australian Government Department of Health, 2020].”

We have also added the following sentence to the caption of Figure 1—figure supplement 1:

“Details on the epidemiological characteristics of locally and overseas acquired infections are available in the Australian Department of Health fortnightly epidemiological reports [National Centre for Immunisation Research and Surveillance, 2020].”

19) Could there be more clarity about the uncertainty presented in the R estimation figures legends? There seems to be a lot of confusion on Twitter etc in interpreting these graphs in terms of variance around the mean/median or variance in the R (some people transmitting more than others), therefore this is an opportunity to state very clearly what the uncertainty represents.

We agree that communicating the meaning of uncertainty to a broader audience is important and thank the reviewer for suggesting we take the opportunity to make some clarifications. The uncertainty in the *R_eff_* estimates represent variability in a population-level average as a result of missing or as yet unreported data, rather than individual-level heterogeneity in transmission (*i.e.*, the variation in the number of secondary cases generated by each case). This is akin to the variation represented by a confidence interval (i.e., variation in the estimate resulting from a finite sample), rather than a prediction interval (*i.e.*, variation associated with an individual observation). Sources of missing or as yet unreported data are described in detail in the Materials and methods section.

We have added the following text to the Materials and methods section:

“The uncertainty in the *R_eff_* estimates (shown in Figure 2, Figure 2—figure supplement 1, Figure 2—figure supplement 2, and Figure 2—figure supplement 3) represents variability in a population-level average as a result of imperfect data, rather than individual-level heterogeneity in transmission (i.e., the variation in the number of secondary cases generated by each case). This is akin to the variation represented by a confidence interval (i.e., variation in the estimate resulting from a finite sample), rather than a prediction interval (i.e., variation in individual observations).”

And we have included the following text in the caption for Figure 2:

“The uncertainty in the *R_eff_* estimates represent variability in a population-level average as a result of imperfect data, rather than individual-level heterogeneity in transmission (i.e., the number of secondary cases generated by each case).”